# Analysis of Cost-Effective Methods to Reduce Industrial Wastewater Emissions in China

**Pengfei Sheng [1], Yaowu Dong [2],\*　and Marek Vochozka [3]**

[1]　School of Economics, Henan University; Kaifeng 475 002, China; sheng_pf@163.com
[2]　School of Public Finance, Guizhou University of Commerce, Guiyang 550 014, China
[3]　The Institute of Technology and Business in Ceske Budejovice, České Budějovice 370 01, Czech Republic; 4017@mail.vstecb.cz
\*　Correspondence: dongyaowu@gzcc.edu.cn

**Abstract:** To reduce industrial wastewater emissions, likely scenarios involve saving water in the production process or treating the emissions that are discharged. In this regard, our paper aims to evaluate the costs of these two paths and then analyze whether the industrial sector has made a good trade-off. In particular, we measured costs of the two paths by shadow prices of water use and wastewater emissions, and then we built a non-parametric input–output model to produce the estimates. For 2015, the shadow price of water use was 37.85 RMB/ton at the national level, which indicated the marginal cost of saving each ton of water was 37.864 RMB and that of wastewater emissions was 141.759 RMB/ton, which meant that the marginal cost of abating each ton of wastewater emissions was 141.759 RMB. Over the period 2004–2015, both shadow prices exhibited an upward trend at the national and regional levels, which suggested there was an increased cost to reduce emissions. However, the two shadow prices did not follow a common trend, but deviated from each other in most of China's provinces, which resulted in a bad trade-off between the two scenarios. As a result, the bad trade-off not only lowered the efficiency to reduce emissions, but it was also linked to a high cost.

**Keywords:** wastewater emissions; shadow price; saving water; cost-effective; abatement cost

## 1. Introduction

Economic activity produces many desirable outputs for consumption and investment, but also generates some undesirable outputs [1], which includes industrial wastewater emissions. Generally, industrial production attempts to produce as many desirable outputs, while generating fewer emissions. However, reducing emissions is not achieved at no cost, and requires extra input, therefore a cost-effective analysis contributes to smart decisions within the industrial sector. In particular, the industrial sector can reduce emissions either by saving water used in the industrial production process or by treating emissions that are discharged. Thus, the industrial sector needs to evaluate the costs of the two paths, and then make a wise trade-off to reduce emissions effectively at low cost. However, this leads to two problems. On the one hand, there is no effective market price for the emissions, and the market price of water would be biased because of imperfect competition. On the other hand, not all industrial sectors perform well in the production frontier and, therefore, they can only make the trade-off at their specific level of production efficiency. More specifically, it is assumed that each industrial sector can make a deal within a perfectly competitive market. Then water's shadow price is measured by its marginal productivity in the industrial production process, while it is workable to use the marginal abatement cost of wastewater emissions to represent the associated shadow price. Therefore, our study aims to evaluate the cost of the two paths by using the shadow price of industrial

water use and industrial wastewater emissions and then to conduct a comparative analysis to decide whether China's industrial sectors can make smart trade-offs at the regional and provincial levels.

From the data published by National Bureau of Statistics of China, China's industrial sector has experienced rapid growth, with an annual average growth rate of 10.34% during 2004–2015. Meanwhile, China's fast-growing industrial sector also produced 19.95 billion tons of industrial wastewater in 2015, and the concentration of chemical oxygen demand (COD) far outweighed the effluent standard and reached 147.10 mg/L. Moreover, industrial wastewater emissions did not decline during 2004–2015. Consequently, China's industrial sector focused more on increasing industrial output than decreasing emissions of industrial wastewater, and current regulations were not effective in reducing these emissions. As a result, China should build relevant policies associated with cost-effective management of reducing emissions, and our work should assist policymakers to achieve dual mandates to reduce emissions and to maintain economic growth.

In this paper, we use the shadow price of water used in the industrial production process to denote the cost to save water, and that of wastewater emissions to represent cost to treat industrial wastewater emissions that are discharged. In detail, results suggest that the shadow price of water used in the industrial production process was 37.864 RMB per ton nationally in 2015, but the shadow price of industrial wastewater emissions was 141.759 RMB per ton. However, these estimates do not mean that the industrial sector can lower the cost of reducing emissions by conserving water, because one ton of water use does not equate to one ton of wastewater emissions. Also, we analyzed the dynamic change in shadow prices of water use and wastewater emissions over the sample period. From 2004 to 2015, both of the shadow prices grew rapidly at the national and regional levels, which demonstrated the increasing cost of reducing emissions. Furthermore, the annual growth rate of the shadow price of water use was correlated negatively with that of wastewater emissions in most of China's provinces. These results mean that these provinces did not make a wise trade-off between the two paths and, therefore, decisions were not effective in reducing the cost of emissions.

Our paper contributes to the literature on this important topic in three ways. First, reducing industrial wastewater emissions is achieved by two paths: either by saving water used in industrial production or by treating emissions that are discharged. Our work aims to analyze industrial sector's choices by evaluating the costs of the two paths. Second, we built a non-parametric input–output model that includes water use and wastewater emissions, and then we used the shadow price of water use and wastewater emissions to denote the costs of the two paths. Third, we investigated how the dynamic change in the shadow price of water use correlated with that of wastewater emissions to evaluate whether industrial sectors made a good trade-off in the long-term. Although many studies have discussed the efficiency of reducing emissions in the production process, few studies have addressed how much reducing emissions costs and whether industrial sector makes a better trade-off, which is the main contributions of this paper.

## 2. Literature Review

This paper investigated the cost of reducing emissions, and we reviewed previous studies that addressed the effects of reducing emissions to economic growth. Generally, previous studies produced mixed or inconclusive results. Some concluded that emission reduction policies had significant negative effects on economic growth [2,3]. Whereas, others believed that such policies improved overall economic performance [4–11]. Nevertheless, these inconsistent results were due in part to different types of environmental policies that caused different effects on economic performance. As a result, there should be feasible polices to reduce emissions with less damage to economic performance and, therefore, it is necessary to evaluate environmental policies from a cost-effective perspective [12].

The cost-effective analysis of environmental policies involves how to measure the costs thereof. There is no competitive market for the emissions of pollutants. As a consequence, the costs of pollution reduction have always been measured using a shadow price [13]. There are two practical methods for estimating the shadow price. Some researchers used a production function [14,15] or a cost function [16]

to estimate the relative shadow price. Other researchers have modeled the production process with a distance function [17,18] and estimated the shadow price using the method of data envelopment (DEA).

Using these methods, there is a large quantity of studies that have contributed to the estimation of the costs of emission pollutants. With a dataset of 51 coal-fired power plants that operated during 1977–1986, Lee [19] found that the overall weighted average shadow price of sulfur dioxide emissions was 0.076 per pound at the constant 1997 price in USD. Färe et al. [13] used the directional output distance function to derive shadow prices of polluting outputs and found that the costs of pollution accounted for approximately 6% of the crop and animal revenues in the US agricultural sector during 1960–1996. Mekaroonreung et al. [20] conducted a convex non-parametric least squares approach to estimate shadow prices of sulfur dioxide and nitric oxide of US coal-powered plants, and their results suggested that it cost 201–345 USD per ton for sulfur dioxide emissions and 409–1352 USD per ton for nitric oxide. Wei et al. [21] utilized the dataset of China's 124 thermal power enterprises from 2004 and found that the average shadow price of carbon dioxide was 316.89 USD per ton. Ambrey et al. [22] developed a micro-econometric life satisfaction model to derive the shadow price of air pollution, which was measured by the value of willingness-to-pay, and their results yielded a shadow price for pollution reduction of approximately 5000 AUD in terms of annual household income in the region of south-east Queensland, Australia. Using a sample of 25 wastewater treatment plants in Spain in 2010, Molinos-Senante et al. [23] estimated that the shadow price of carbon dioxide emissions comprised 17.7% of the cost of treated water on average. Tang et al. [24] used a parameterized quadratic form of the direction distance function to construct a data set of China's 30 provinces during 2001–2010; their results showed that the average shadow price of chemical oxygen demand (COD), total nitrogen (TN), and total phosphorus (TP) were 8266 Yuan per ton, 25,560 Yuan per ton, and 10,160 Yuan per ton, respectively, for the entire country. Wang et al. [25] used a global and non-radial and directional distance function to measure the shadow price of industrial water, and their results implied that the actual prices of industrial water were much lower than its shadow price during 2004–2012 in China. Zeng et al. [26] applied the out directional distance function to produce a shadow price of the emissions of sulfur dioxide and found that the average shadow price exhibited a fluctuating downward trend for the period 2001–2012 in China. This body of literature represents useful research on the measurement of emission pollutants that can provide valuable references for the formation of environmental policies.

## 3. Methods

We were interested in how to reduce emissions of industrial wastewater with a cost-effective option. Conventional wisdom suggested that industrial wastewater emissions could be reduced either by saving water or by treating the discharged emissions. Therefore, a decision-making unit (DMU) requires making a trade-off between two paths in the short-term, whereby the path with the lower cost is preferable. Following the law of increasing marginal costs, costs of the two paths could follow the same trend when the DMU makes a good trade-off.

As there was no efficient market for the two paths, prices were not available; therefore it was essential to model a competitive market to reduce emissions. First, we assumed the process of industrial production as in Equation (1), which shows that every DMU makes use of capital ($K$), labor ($L$), energy ($E$), and water ($W$) to produce the desirable output ($Y$), and it also generates an undesirable output that is denoted by the emissions of industrial wastewater ($B$).

$$P(K_i, L_i, W_i) = \{(Y_i, B_i) : \ (K_i, L_i, E_i, W_i) \ can \ produce \ (Y_i, B_i)\} \tag{1}$$

As suggested by Chung et al. [27], the above production equation must satisfy the following constraints: (1) undesirable output is weakly disposable, which means either inputs must be diverted or desirable outputs must be cut back to reduce the undesirable output; (2) desirable outputs are strongly disposable, which indicates that the undesirable output cannot be increased continuously because

inputs are fixed; and (3) desirable outputs and undesirable outputs are "null-jointly", which means that the desirable output could be reduced to zero if no undesirable output was generated.

The model $P(K_i, L_i, E_i, W_i)$ assumes that each DMU performs at the same level of technology, but the productive efficiency might be heterogeneous. Thus, we used the method of the directional distance function [27] to obtain an efficient path whereby each DMU performed on the production frontier.

$$D(K_i, L_i, W_i) = \sup \left\{ \begin{array}{l} (SY_i, SB_i, SK_i, SL_i, SE_i, SW_i) : \\ (Y_i + SY_i, B_i - SB_i, K_i - SK_i, L_i - SL_i, E_i - SE_i, W_i - SW_i) \end{array} \right\} \quad (2)$$

where $SY$, $SB$, $SK$, $SL$, $SE$, and $SW$ denote slacks for desirable output, undesirable output, capital input, labor input, energy, and water, respectively. Equation (2) means the $i^{th}$ DMU can increase as much desirable output while decrease as the many undesirable outputs and inputs. When $D(K_i, L_i, E_i, W_i)$ is equal to zero, the $i^{th}$ DMU has no efficiency loss for both the production of the undesirable output and pollution reduction.

From the Equation (2), when there is no efficiency loss for the $i^{th}$ DMU, the input sets and the output sets can be adjusted as follows:

$$\begin{array}{l} Y_i^* = Y_i + SY_i; B_i^* = B_i - SB_i; \\ K_i^* = K_i - SK_i, L_i^* = L_i - SL_i, E_i^* = E_i - SE_i, W_i^* = W_i - SW_i \\ D(K_i^*, L_i^*, E_i^*, W_i^*) = 0 \end{array} \quad (3)$$

Equation (3) shows the path to achieve the production frontier for the $i^{th}$ DMU. Therefore, we can define the efficiency of reducing emissions and for saving water. Emissions reduction efficiency (*ERE*) is measured by the ratio of optimal emissions to actual emissions, whereas water efficiency (*WUE*) is measured by the ratio of optimal water use to actual use in industrial production.

$$\begin{array}{l} ERE = {}^{B_i^*}/_{B_i} = 1 - {}^{SB_i}/_{B_i}; \\ WUE = {}^{W_i^*}/_{W_i} = 1 - {}^{SW_i}/_{B_i}; \end{array} \quad (4)$$

Following the efficient path in Equation (3), the DMU can maximize its profit by the following solution:

$$\begin{array}{ll} \max & Z_i = P_i^Y Y_i^* - P_i^B B_i^* - P_i^W W_i^* - P_i^E E_i^* - P_i^K K_i^* - P_i^L L_i^* \\ subject\ to: & D(K_i^*, L_i^*, E_i^*, W_i^*) = 0 \end{array} \quad (5)$$

where $P_i^Y$, $P_i^B$, $P_i^W$, $P_i^E$, $P_i^K$, $P_i^L$ describe the prices for the desirable output, undesirable output, water, energy capital input, and labor input, respectively. $Z_i$ denotes the total profits derived from industrial production and emissions reduction, which shows the target of the $i^{th}$ DMU.

With the Lagrange multiplier method, Equation (5) can be rewritten as described in Equation (6) and the $\lambda$ represents the Lagrange multiplier.

$$\max \quad F = P_i^Y Y_i^* - P_i^B B_i^* - P_i^W W_i^* - P_i^E E_i^* - P_i^K K_i^* - P_i^L L_i^* + \lambda D(K_i^*, L_i^*, E_i^*, W_i^*) \quad (6)$$

With regarding to Equation (6), its first order conditions are listed as the following. In detail, Equation (7) corresponds to the desirable output, the undesirable output and the four kinds of input, and Equation (8) implies that the $i^{th}$ DMU produces on the production frontier.

$$\begin{array}{ll} \frac{\partial F}{\partial Y_i^*} = P_i^Y + \lambda \frac{\partial D(K_i^*, L_i^*, E_i^*, W_i^*)}{\partial Y_i^*} = 0; & \frac{\partial F}{\partial B_i^*} = -P_i^B + \lambda \frac{\partial D(K_i^*, L_i^*, E_i^*, W_i^*)}{\partial B_i^*} = 0 \\ \frac{\partial F}{\partial W_i^*} = -P_i^W + \lambda \frac{\partial D(K_i^*, L_i^*, E_i^*, W_i^*)}{\partial W_i^*} = 0; & \frac{\partial F}{\partial E_i^*} = -P_i^E + \lambda \frac{\partial D(K_i^*, L_i^*, E_i^*, W_i^*)}{\partial E_i^*} = 0; \\ \frac{\partial F}{\partial K_i^*} = -P_i^K + \lambda \frac{\partial D(K_i^*, L_i^*, E_i^*, W_i^*)}{\partial K_i^*} = 0; & \frac{\partial F}{\partial L_i^*} = -P_i^L + \lambda \frac{\partial D(K_i^*, L_i^*, E_i^*, W_i^*)}{\partial L_i^*} = 0 \end{array} \quad (7)$$

$$\frac{\partial F}{\partial \lambda} = D(K_i^*, L_i^*, E_i^*, W_i^*) = 0 \tag{8}$$

As desirable output is measured by the industrial value added, it is workable to set the shadow price of the desirable output to be 1, and then shadow prices for water and wastewater emissions can be defined as outlined in Equation (9)

$$\begin{aligned} P_i^B &= -\left( \frac{\partial D(K_i^*, L_i^*, E_i^*, W_i^*)}{\partial B_i^*} \Big/ \frac{\partial D(K_i^*, L_i^*, E_i^*, W_i^*)}{\partial Y_i^*} \right) \\ P_i^W &= -\left( \frac{\partial D(K_i^*, L_i^*, E_i^*, W_i^*)}{\partial W_i^*} \Big/ \frac{\partial D(K_i^*, L_i^*, E_i^*, W_i^*)}{\partial Y_i^*} \right) \end{aligned} \tag{9}$$

As the behavior of DMUs is assumed to be efficient in Equation (3), the given market is consistent with a perfectly competitive market. As a result, shadow price of wastewater emissions is equivalent to its marginal abatement cost (MAC), which the DMU must take to treat each unit of the emission. Meanwhile, shadow price of water denotes its marginal productivity (MPW), indicating how much each unit of water contributes to the desirable output. Remarkably, Equation (1) cannot describe how industrial sectors convert water into emissions of industrial wastewater, therefore a higher value of *MAE* than *MPW* does not mean that saving water is the better option to reduce emissions, and vice versa. Thus, it is workable to investigate whether the *MAE* and the *MPW* follow the same trend, and then we can conclude whether DMU make a good trade-off to reduce the emissions

To solve Equations (2) and (5), it is practicable to use the data envelopment [28], which is convenient to evaluate the production model with multiple inputs and multiple outputs. Meanwhile, it is practicable to assume no significant technology progress happens in the short-term, and then we can evaluate the two equations globally over the sample period [29]. Following suggestions provided by Tone [30] and Fukuyama et al. [31], the solution takes the following form:

$$\begin{aligned} D(K_{it}, L_{it}, E_{it}, W_{it}) = \max &\frac{\frac{1}{2}(SY_{it}/Y_{it} + SB_{it}/B_{it}) + \frac{1}{4}(SK_{it}/K_{it} + SL_{it}/L_{it} + SE_{it}/E_{it} + SW_{it}/W_{it})}{2} \\ &\sum_{t=1}^{T} \sum_{j=1}^{N} z_{jt} K_{jt} + SK_{jt} = K_{it}; \sum_{t=1}^{T} \sum_{j=1}^{N} z_{jt} L_{jt} + SL_{jt} = L_{it}; \sum_{t=1}^{T} \sum_{j=1}^{N} z_{jt} W_{jt} + SW_{jt} = W_{it} \\ &\sum_{t=1}^{T} \sum_{j=1}^{N} z_{jt} E_{jt} + SE_{jt} = E_{it}; \sum_{t=1}^{T} \sum_{j=1}^{N} z_{jt} Y_{jt} - SY_{jt} = Y_{it}; \sum_{t=1}^{T} \sum_{j=1}^{N} z_{jt} B_{jt} + SB_{jt} = B_{it}; \\ &\sum_{t=1}^{T} \sum_{j=1}^{N} z_{jt} = 1; j = 1, \cdots N; t = 1, \cdots N \end{aligned} \tag{10}$$

## 4. Data

### 4.1. Data Description

Our study attempts to measure the marginal abatement cost of treating industrial wastewater emissions and marginal productivity of water used in industrial production in China and to find a cost-effective path to reduce emissions in the industrial sector. We used a provincial dataset for China for 2004–2015, which excluded Tibet, Hong Kong, Macau, and Taiwan because of incomplete information. The data set was obtained from the "*China Environment Yearbook*", the "*China Industry Economy Statistical Yearbook*", and the "*China Statistical Yearbook*", within the *National Bureau of Statistics of People's Republic of China*.

Our input–output model used labor, capital stock, energy, and water as the input, and we treated industrial value added as the desirable output and industrial wastewater emissions as the undesirable output. Industrial value added was calculated at the constant price for 2004. Labor was measured by the average number of industrial employees at the beginning and at the end of the year. To calculate capital stock, we required four variables, which included capital stock in the base year, annual stock flow, depreciation rate, and the associated price index. In detail, we used the fixed assets-net value of all industrial enterprises for 2004 as the capital stock of the base year, and the data were collected from *Communique on Major Data of the First National Economic Census of China* (2004). We estimated the annual

stock flow by the total investment in fixed assets in the industrial sector, denoted the price index using the price index for investments in fixed assets, and measured the depreciation rate by the depreciation rate for industrial enterprises above a designated size. Energy was estimated using the total energy consumption of the industrial sector. Water referred to the water volume that industrial enterprises used to manufacture, machine, cool, wash and so on, but excluded the recycled water. Following the *Integrated Wastewater Discharge Standard of China* (1998), Wastewater emissions contained the higher pollutants concentration than the standard and were emitted by any industrial enterprises' sewage outfall. In detail, pollutant concentrations of wastewater emissions are higher than 50 mg/L for the COD, and 1.5 mg/L for the ammonia nitrogen, and 10 mg/L for the total nitrogen, and 0.3 mg/L for the total phosphorus, and 1.0 mg/L for the petroleum, and so on. Water use and wastewater emissions were collected from the "*China Environment Yearbook*". Table 1 outlines the descriptive statistics for all variables.

**Table 1.** Descriptive statistics of variables.

| Variables | Units | Maximum | Minimum | Mean | Median | Standard Deviation |
|---|---|---|---|---|---|---|
| Capital Stock | 100 million RMB | 63,738.538 | 290.152 | 10,747.453 | 7356.721 | 0.991 |
| Labor | 10,000 persons | 2362.440 | 29.920 | 586.243 | 350.595 | 0.923 |
| Water | 100 million ton | 239.000 | 2.390 | 45.734 | 27.955 | 0.978 |
| Energy | 10,000 ton | 24,574.000 | 424.000 | 7842.911 | 6023.600 | 0.679 |
| Industrial Added Value | 100 million RMB | 26,569.970 | 142.330 | 4800.925 | 3318.605 | 1.004 |
| Wastewater Emissions | 100 million ton | 29.632 | 0.354 | 7.586 | 4.883 | 0.843 |

## 4.2. Industrial Water Use and Industrial Wastewater Emissions in China

China's emissions of industrial wastewater reached a peak of 24.66 billion tons in 2007, and showed a downward trend with an annual decreasing rate of 2.6% from 2007 to 2014 (Figure 1). Industrial water use reached a peak of 146 billion tons in 2011, and it showed a slightly downward trend from 2011 to 2015. Moreover, China's industrial water use in 2015 was much higher than in 2003. As a result, we concluded that China did much to reduce the emissions of industrial wastewater during 2003–2015, but the reduction was mainly due to cleaning the emissions rather than using less water.

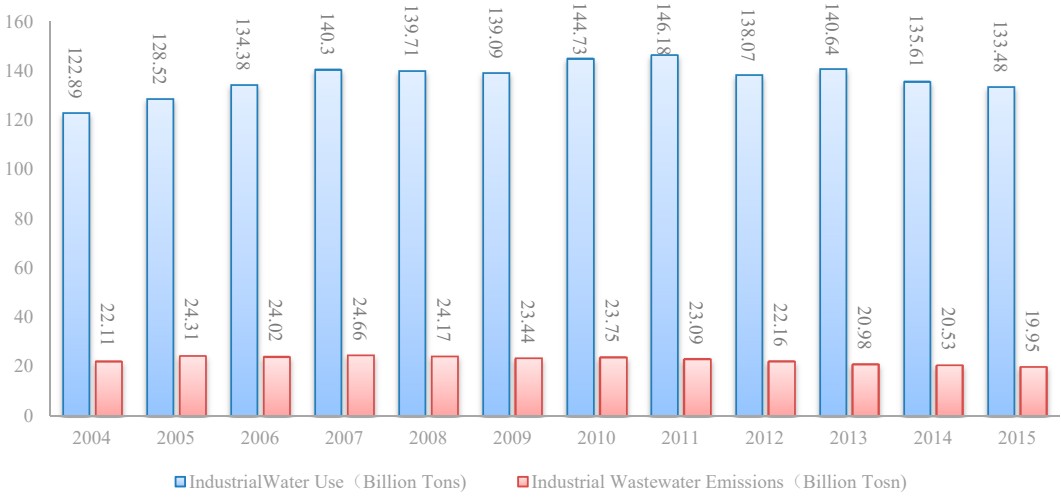

**Figure 1.** Industrial water use and industrial wastewater emissions in China.

## 5. Results

### 5.1. Shadow Prices of Water Use and Wastewater Emissions

Because environmental issues are always associated with regional economic development, we discuss the shadow prices of water use and wastewater emissions across different regions of China.

In terms of social-economic status and geographical location, China is divided into three regions: the eastern region, the central region, and the western region. The eastern region includes Beijing, Tianjin, Hebei, Liaoning, Shanghai, Jiangsu, Zhejiang, Fujian, Shandong, Guangdong, and Hainan, which link people of high social- economic status located in the eastern coastal areas of China. The central region comprises Shanxi, Jinlin, Heilongjiang, Anhui, Jiangxi, Henan, Hubei, and Hunan, which include people who enjoy a modest level of economic development located in the central areas of China. The western region is the most underdeveloped, and includes Inner Mongolia, Guangxi, Chongqing, Sichuan, Guizhou, Yunnan, Shannxi, Gansu, Qinghai, Ningxia, Xinjiang, and Tibet (excluded from our sample).

From the figure (2), In 2015, the shadow prices of water use indicated that to reduce each ton of water used in industrial production would cost 37.864 RMB/ton, 49.227 RMB/ton, 27.819 RMB/ton, and 27.793 RMB/ton for all of China, the eastern region, the central region, and the western region, respectively (Figure 2). Relevant prices for water use and wastewater emissions were calculated by the average values weighted by provincial amounts of water use and wastewater emissions, respectively. Over the sample period, all three regions enjoyed great progress in the shadow price, with annual growth rates of 15.77% for the eastern region, 16.05% for the central region, and 12.667% for the western region. Likewise, the shadow price for all of China also experienced a rapid annual growth rate of 15.23%. Furthermore, the eastern region had the highest shadow price, which was 76.95% higher than the central region and 77.12% higher than the western region. In detail, the rising shadow prices indicated that China's industrial sector improved their environmental awareness and tried to increase the marginal productivity of water, which resulted in less water use in the production process. Meanwhile, regional differences in the shadow price implied that China's industrial sector did not achieve the optimal allocation of water resources, which argued that water conservation and reduction of wastewater emissions were not sufficient.

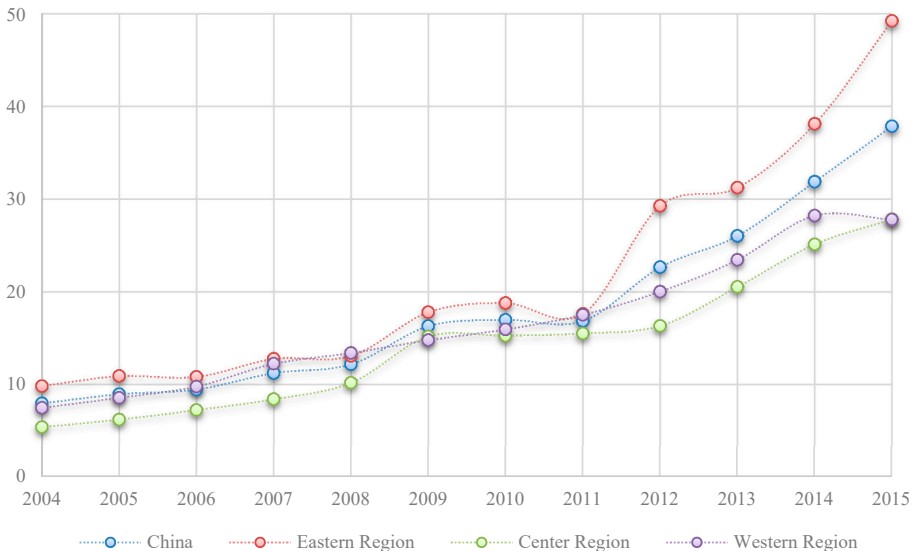

**Figure 2.** Shadow price of water use in China by region. Unit: RMB/ton.

Figure 3 reported the shadow price of wastewater emissions across China and the three regions. Results suggested that to reduce industrial wastewater emission, it would cost 141.759 RMB per ton for all of China, 157.917 RMB for the eastern region, 125.594 RMB for the central region, and 121.136 RMB for the western region, in 2015. Meanwhile, all three regions saw rapid growth in shadow prices, with annual growth rates of 12.21% for the eastern region, 13.80% for the central region, and 16.32% for the western region; increased shadow prices implied that the regions encountered growing costs to reduce industrial wastewater emissions. Over the sample period, the eastern region had the highest shadow price for wastewater emissions, followed by the central region and the western region.

However, differences among the three regions narrowed from 2004 to 2011 but widened from 2011 to 2015. In detail, the shadow price of the eastern region was 25.74% higher than the central region and 30.36% higher than the western region. Regional differences were notable in that the well-developed eastern region had the highest cost for reducing industrial wastewater emissions, with the result that polluting industries tended to shift from the eastern region to the central and western regions, which depressed the estimate for emissions reduction for China as a whole.

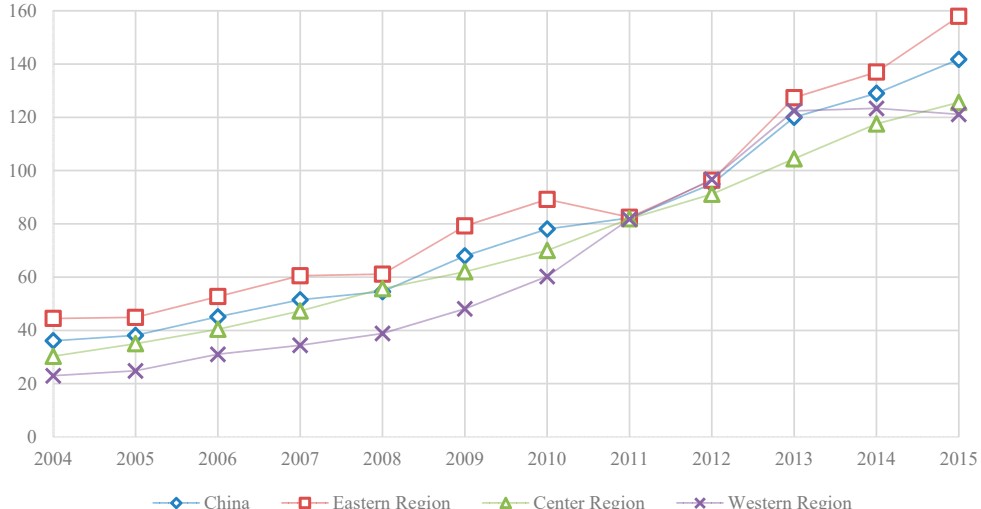

**Figure 3.** Shadow price of wastewater emissions in China by region. Unit: RMB/ton.

In addition to looking at the regions and China as a whole, shadow prices across China's provinces revealed much information about reducing industrial wastewater emissions (Table 2). In 2015, Shandong, Beijing, and Tianjin scored in the top three on the shadow price of water use, which indicated that water resources contributed much to the industrial production of these provinces. Meanwhile, because industrial sector in the three provinces relied much on water use, it was difficult to reduce emissions by lowering water use. On the contrary, Jiangsu, Inner Mongolia, and Shanghai ranked in the bottom three in the shadow price of water use, which indicated that these provinces relied less on water resources for their industrial production. In particular, Shanghai and Inner Mongolia achieved the lowest water use, which showed that their industries relied less on water. Water efficiency was only 0.094 for Jiangsu, which was evidence of bad allocation of water resources in industrial production. Over the sample period, the shadow price of water for Hebei (28.77%), Jiangxi (22.02%), Beijing (21.86%), and Shandong (20.49) grew by >20% a year, while Shanghai and Inner Mongolia exhibited a negative growth rate. As a result, there was a great dispersion in the shadow price among provinces, which were associated closely with the incomplete market for water resources and, thus, the dispersion impeded the progress in water conservation and wastewater reduction.

The shadow price of wastewater emissions in Tianjin, Beijing, and Guangdong were highest at 443.192 RMB/ton, 442.537 RMB/ton, and 205.707 RMB/ton, respectively, which indicated that it was costly for these provinces to reduce wastewater emissions. In contrast, Ningxia, Hainan, and Guizhou had the lowest shadow prices of wastewater emissions of 47.719 RMB/ton, 84.478 RMB/ton, and 85.285 RMB/ton, respectively. Meanwhile, these provinces also showed low values of efficiency for wastewater emissions of 0.108, 0.288, and 0.192, respectively, which suggested that these provinces could mitigate emissions by improving efficiency. Meanwhile, annual growth rate of the shadow price of wastewater emissions also saw great differences among provinces over the sample period. Chongqing, Guangxi, and Sichuan experienced the top three growth rates in shadow prices of 25.60%, 22.87%, and 21.66%, respectively, which implied that these provinces encountered rising costs to reduce emissions. However, Shanghai, Inner Mongolia, and Qinghai had annual growth rates of <6%. In particular, Shanghai and Inner Mongolia were most successful in reducing emissions, which indicated they employed an optimal

allocation in the process to reduce emissions. Whereas, Qinghai had a low efficiency of 0.205 in reducing emissions, which meant that the lower marginal cost resulted in insufficient abatement of pollution.

**Table 2.** Shadow price of water use compared with shadow price of wastewater emissions across China's provinces.

| Provinces | Shadow Price of Water Use (RMB/ton) | | | | Shadow Price of Wastewater Emissions (RMB/ton) | | | |
|---|---|---|---|---|---|---|---|---|
| | 2005 | 2010 | 2015 | Annual Growth Rate | 2005 | 2010 | 2015 | Annual Growth Rate |
| Beijing | 33.273 | 42.327 | 251.392 | 21.86% | 148.201 | 353.445 | 442.537 | 11.31% |
| Tianjin | 54.467 | 241.608 | 79.327 | 3.86% | 67.727 | 233.276 | 443.192 | 17.28% |
| Hebei | 18.753 | 56.835 | 247.476 | 28.77% | 39.493 | 78.280 | 141.601 | 14.09% |
| Liaoning | 53.048 | 42.518 | 59.070 | 9.37% | 40.679 | 129.494 | 156.370 | 13.06% |
| Shanghai | 4.779 | 5.658 | 4.425 | −0.60% | 87.134 | 193.503 | 121.796 | 4.94% |
| Jiangsu | 5.198 | 11.993 | 13.471 | 10.29% | 33.759 | 72.310 | 145.710 | 14.62% |
| Zhejiang | 13.826 | 25.360 | 40.594 | 12.13% | 36.054 | 55.302 | 113.196 | 10.71% |
| Fujian | 4.787 | 8.847 | 18.050 | 14.12% | 25.590 | 55.410 | 133.419 | 16.10% |
| Shandong | 59.296 | 69.078 | 470.731 | 20.49% | 68.526 | 87.845 | 155.309 | 8.60% |
| Guangdong | 6.537 | 10.412 | 14.761 | 9.28% | 50.430 | 120.730 | 205.707 | 11.70% |
| Hainan | 11.768 | 14.576 | 32.794 | 12.07% | 29.799 | 74.049 | 84.478 | 10.72% |
| Shanxi | 23.699 | 49.812 | 83.237 | 13.38% | 63.344 | 70.164 | 115.786 | 6.96% |
| Jilin | 12.547 | 21.929 | 30.214 | 8.96% | 36.239 | 91.147 | 148.855 | 12.76% |
| Heilongjiang | 4.617 | 8.009 | 31.502 | 19.10% | 65.867 | 136.035 | 193.228 | 11.55% |
| Anhui | 4.302 | 10.869 | 19.575 | 16.25% | 33.167 | 70.853 | 127.043 | 14.97% |
| Jiangxi | 3.907 | 13.757 | 24.895 | 22.02% | 28.581 | 51.800 | 85.552 | 12.69% |
| Henan | 10.765 | 46.121 | 65.885 | 19.33% | 39.790 | 68.329 | 126.010 | 12.37% |
| Hubei | 5.976 | 7.491 | 16.044 | 10.65% | 28.981 | 62.536 | 127.851 | 16.64% |
| Hunan | 3.575 | 9.668 | 18.382 | 17.90% | 19.273 | 57.669 | 122.481 | 19.78% |
| Inner Mongolia | 15.849 | 20.131 | 8.937 | −4.57% | 66.626 | 124.160 | 93.982 | 5.40% |
| Guangxi | 4.259 | 11.598 | 20.440 | 17.14% | 9.672 | 20.904 | 92.929 | 22.87% |
| Chongqing | 4.359 | 7.435 | 19.435 | 16.63% | 18.039 | 84.134 | 198.604 | 25.60% |
| Sichuan | 7.597 | 15.134 | 29.670 | 15.18% | 23.378 | 75.251 | 173.066 | 21.65% |
| Guizhou | 6.133 | 9.980 | 24.355 | 15.01% | 50.642 | 94.437 | 85.285 | 6.89% |
| Yunnan | 10.806 | 17.951 | 35.445 | 13.00% | 38.785 | 80.793 | 87.744 | 9.79% |
| Shannxi | 20.295 | 45.011 | 41.909 | 7.43% | 39.036 | 77.920 | 161.426 | 13.82% |
| Gansu | 10.542 | 22.494 | 48.870 | 15.30% | 43.354 | 86.751 | 107.604 | 10.86% |
| Qinghai | 11.364 | 40.075 | 74.173 | 18.00% | 28.769 | 53.413 | 90.961 | 5.55% |
| Ningxia | 17.929 | 34.699 | 46.102 | 10.08% | 11.398 | 21.891 | 47.719 | 7.22% |
| Xinjiang | 19.416 | 29.903 | 43.201 | 9.60% | 45.178 | 63.007 | 88.076 | 6.61% |

Note: all values are at constant 2002 price.

## *5.2. Discussion*

The industrial sector can achieve the final reduction in industrial wastewater emissions, either by saving water or by treating emissions in the industrial production process. If the industrial sector tries to reduce emissions by reducing water use, the associated water efficiency would increase. Likewise, treating more emissions is also linked to the rising efficiency in wastewater emissions. In 2004, China's water efficiency was 0.246, which indicated that 75.4% of industrial water use could be reduced when every province performed at the production frontier (Figure 4). Similarly, the efficiency of wastewater emissions was 0.197, which meant that 80.3% of total emissions could be treated when all provinces attain the production frontier. Since 2010, China's water use efficiency has been higher than wastewater emissions efficiency; that is, China's provinces accomplished more in water conservation than in treatment of wastewater emissions. However, with the increasing attention on wastewater emissions, efficiency in reducing emissions has grown faster than that in conserving water use over the entire sample. Notwithstanding, although more emissions were mitigated either by water conservation or emissions treatment, the growing gap between the two paths led to higher costs for emissions abatement.

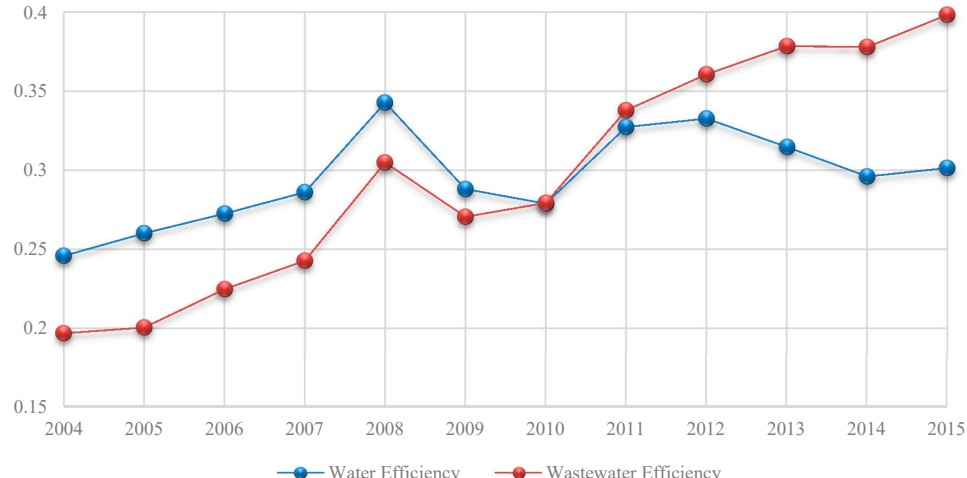

**Figure 4.** Water efficiency compared with wastewater emissions efficiency in China during 2004–2015.

Generally, rational units would make trade-offs between the costs of the two paths in the long-term, and then the shadow price of water use would tend to follow the same trend as that of wastewater emissions. However, information asymmetry and asset specificity will result in suboptimal choices for reducing emissions in the short-term. Indeed, China's industrial sectors have put more effort into water conservation since 2010, however, they have invested more in treating emissions after 2010 (Figure 4). Therefore, we measured the correlation coefficient between the annual growth rates of shadow prices of water use and wastewater emissions for China's 30 provinces to determine whether the two paths followed a common trend. Guangdong, Inner Mongolia, and Shanghai had high correlation coefficients >0.5, which suggested that these provinces demonstrated better trade-offs between the two paths (Figure 5). Yet, correlation coefficients were negative for 18 provinces, which indicated that they made unfavorable decisions. In addition, Hainan and Guangxi had correlation coefficients <−0.6; this indicated that the two provinces reduced emissions by one of the two paths that resulted in low efficiency and relatively high cost of emission reduction. Therefore, most of China's provinces reduced emissions by one of the two paths, and our results suggested that a better trade-off between the two paths might yield a lower cost.

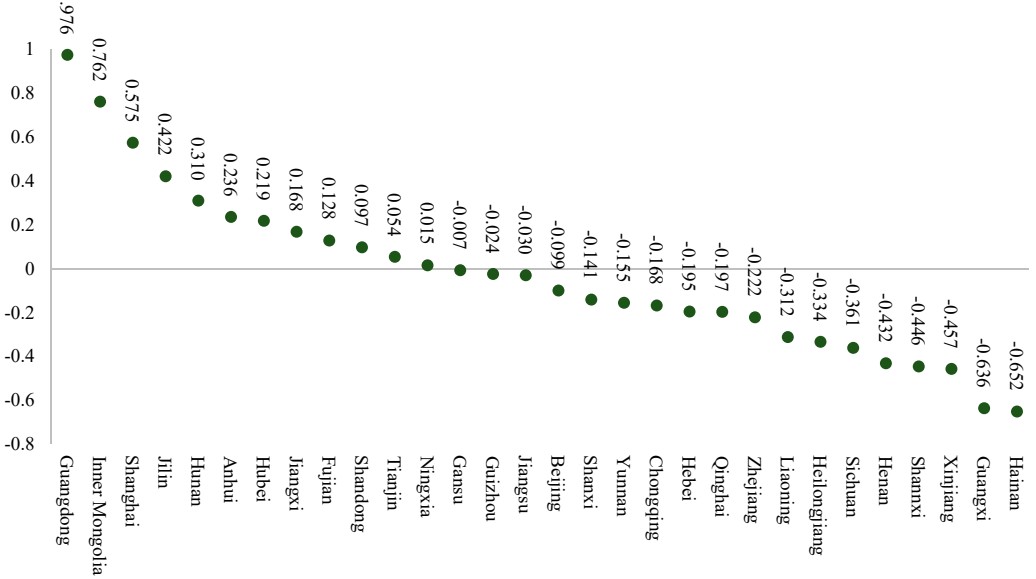

**Figure 5.** Correlation coefficients between annual growth rates of shadow prices of water and wastewater in the provinces in China during 2004–2015.

## 6. Conclusions

This paper aimed to provide a cost-effective analysis of the reduction of industrial wastewater emissions in China. Despite the fact that many studies have evaluated potential reductions in emissions, few of them provided costs associated with the reduction. Our work found that the industrial sector reduced emissions either by saving the water used in the production process or by treating the industrial wastewater that was discharged. Then, it is rational to assume that the industrial sector needed to make trade-offs between the two paths, and the costs of the two paths would follow a common trend. Thus, we built a non-parametric input–output model to estimate the shadow price of water use and wastewater emissions in the industrial production process, and we noted the costs of the two paths.

We used China's provincial dataset for the period 2004–2005, and we used the data envelopment method to produce our estimates. In 2015, the shadow price of water use was 37.864 RMB/ton, which indicated that to save one ton of water cost 37.864 RMB in China; the shadow price of wastewater emissions was 141.759 RMB/ton, which showed that to treat one ton of emissions cost 141.759 RMB. The efficiency in treating wastewater emissions was higher than that of conserving water in 2015. That is, the industrial sector reduced emissions by treating more emissions, but doing so was linked to higher costs. Over the sample period, the two shadow prices followed an upward trend, which suggested that costs to reduce industrial wastewater emissions in China increased. We then measured the correlation coefficients of the annual growth rate of water use and wastewater emissions. Only three provinces had a coefficient >0.5, but 18 provinces had negative coefficients, which indicated that most provinces made a bad trade-off between the two paths. These trade-offs not only worked against reducing emissions of industrial wastewater, but also led to higher costs to reduce emissions.

The potential explanations as to why most of China's provinces do not make a better trade-off between the two paths over the past decade were due to the following three reasons, which are not mutually exclusive. First, there was no perfect information on costs of the two paths before the industrial sector decided how to reduce emissions, and then the asymmetric information led to suboptimal resolutions. Second, water conservation and wastewater treatment required considerable investment in specific assets, but the industrial sector could not find an alternative path in the short -term. Finally, local governments corner the water market and have more at promoting industrial growth, which enabled the industrial sector to acquire water at a lower price than water's marginal productivity. Moreover, investigating how these factors affected the industrial sector's choice would require more detailed data and was beyond the subject of this research.

From the perspective of reducing emissions, we found three important implications for policy making. (1) The industrial sector can reduce wastewater emissions either by saving water use, or by treating wastewater emissions, and then policies aiming at reducing the emissions should consider the two kinds of paths as a whole. (2) Our estimates on water use and wastewater emissions over the past decade provide meaningful references for public policies that aim to lower the costs of emissions abatement. (3) Introducing marketing mechanisms to water supply can alleviate the information asymmetry, and then the industrial sector could make a better trade-off to reduce emissions.

**Author Contributions:** Conceptualization, P.S.; methodology, P.S. and Y.D.; software, Y.D.; validation, M.V.; formal analysis, P.S.; investigation, P.S. and Y.D.; resources, P.S. and Y.D.; data curation, M.V.; writing—original draft preparation, P.S. and Y.D.; writing—review and editing, M.V.; visualization, M.V.; supervision, P.S.; project administration, P.S.; funding acquisition, P.S. and Y.D. All authors have read and agreed to the published version of the manuscript.

**Funding:** This research was funded by Talent Program of Universities of Henan Province: 20HASTIT032, the Doctoral Research Project of Guizhou University of Commerce: BSKY2018018

**Conflicts of Interest:** The authors declare no conflict of interest.

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
