# Peer review of "Analysis of Cost-Effective Methods to Reduce Industrial Wastewater Emissions in China"

_water, doi:10.3390/w12061600_

Round 1

Reviewer 1 Report

Overall, a good paper and information worthwhile for distribution.  A few suggestions/comments/questions as following:

  1. In the Abstract, please make sure that you add the units after the numbers, 37.85 and 141.759; and add either appropriate time or quantity units for the other two RMB values, i.e., either RMB/Yr or RMB/Ton etc.
  2. In the Introduction, please add a statement or two explaining what you mean by "shadow price" and present a list of assumptions you have made for determining shadow price of both the water use and pollution.
  3. It would be helpful to readers if you present typical range of water pollution either as ppm or % you observed in your study, present this information both in Introduction and in Data sections.
  4. Methods is the hardest part of your paper to understand, you may want to explain your equations 2-9 in simple language.
  5. In Table 1, you need to add Units for each number.
  6. In Figure 1, consider using unit of Billion Tons instead of 100 Million Tons for wastewater emission, that will make Figure 1 easy to understand.
  7. Table 2 - add units.

Author Response

Dear Reviewers,

Thank you very much for your kind letter and helpful comments, which are highly conducive to improve our paper with the number water-817698. We have thoroughly considered all comments and substantially revised our manuscript, and tried our utmost to address all concerns. As non-native speakers in English, we also invite Thomas A. Gavin, Professor Emeritus, Cornell University, for help with editing this paper. If any questions, please contact us without hesitate. Following is the revision notes for each comment.

Overall, a good paper and information worthwhile for distribution. A few suggestions/ comments/ questions as following:

  1. In the Abstract, please make sure that you add the units after the numbers, 37.85 and 141.759; and add either appropriate time or quantity units for the other two RMB values, i.e., either RMB/Yr or RMB/Ton etc.

Responses: We followed your suggestion and revised the description in the abstract.

  1. In the Introduction, please add a statement or two explaining what you mean by "shadow price" and present a list of assumptions you have made for determining shadow price of both the water use and pollution.

Responses: We addressed the meaning of shadow price for water and wastewater emissions in the first paragraph of the introduction. Please see the following or the introduction of the manuscript.

“More specifically, it is assumed that each industrial sector can make a deal in a perfectly competitive market. And then water’s shadow price is measured by its marginal productivity in the industrial production process, while it is workable to use the marginal abatement cost of wastewater emissions to represent the associated shadow price.”

  1. It would be helpful to readers if you present typical range of water pollution either as ppm or % you observed in your study, present this information both in Introduction and in Data sections.

Responses: We presented typical range of water pollution as ppm observed in the introduction and data section.

“Meanwhile, China’s fast-growing industrial sector also produced 19.95 billion tons of industrial waste water in 2015, and the concentration of chemical oxygen demand (COD) far outweighed the effluent standard and reached 147.10 mg/L.” (Introduction Part)

“Water referred to the water volume that industrial enterprises used to manufacture, machine, cool, wash and so on, but excluded the recycled water. Following the Integrated Wastewater Discharge Standard of China (1998), Wastewater emissions contained the higher pollutants concentration than the standard and were emitted by any industrial enterprises’ sewage outfall. In detail, pollutants concentration of wastewater emissions are higher than 50 mg/L for the COD,  and 1.5 mg/L for the ammonia nitrogen, and 10 mg/L for the total nitrogen, and 0.3 mg/L for the total phosphorus, and 1.0 mg/L for the petroleum, and so on.” (Data part)

  1. Methods are the hardest part of your paper to understand, you may want to explain your equations 2-9 in simple language.

Responses: we added more explanation for the equations 2-9 in the manuscript.

  1. In Table 1, you need to add Units for each number.

Responses: We added the units for each variable in table 1.

  1. In Figure 1, consider using unit of Billion Tons instead of 100 Million Tons for wastewater emission that will make Figure 1 easy to understand.

Responses: we used the unit of billion tons for wastewater emissions in the figure 1.

  1. Table 2 - add units.

Responses: we added the units in table 2.

Reviewer 2 Report

The authors set out to examine two paths firms can use in order to reduce water pollution from their operations.  The paths are a) reduce the amount of water used in the production process and b) treat the emissions flowing out of their facilities.  This is an important topic and one that is appropriate for this journal.

The authors include an introduction section where they lay out the framework and organization of their manuscript.  In this section they also explain how this paper adds to the literature on the topic at hand.  A key concept they lay out in this section is the concept of shadow prices, which is very important in environmental economics where amenities are not directly bought and sold on a market, and therefore lack explicit, observable prices, but have costs and benefits nonetheless and can therefore only be analyzed economically via shadow prices.

The next section contains an extensive literature review. 

In section 3 the authors go into method.  I really like the model the authors employ here, which is a constrained optimization problem using a LaGrangian equation.  This is highly appropriate.  I do want to say however, that there is a typographical error which must be corrected.  In the first order conditions, the set of equations listed as (7) the second equation in the right hand column is listed as partial F over partial W, which is a repeat of the equation in the left hand column.  That equation is meant to read partial F over partial E.  Please replace that W with the E.  Other than that, the conditions are all correctly derived.

The authors include a brief data section which is followed by the results.

In the results section, in the text, the authors present 2015 shadow prices on water reduction per ton in various regions of China.  They also use figures and a table to show how the shadow prices have changed over the period 2004-2015.  They included a discussion section which focused on efficiency.  

In the conclusion, the authors do a good job of summarizing the implications of their work.  However, the sentence in the final paragraph on implications that begins with 1) is not clear.  It reads: “public policies that aim to save water resources closely related to emissions reduction, and then the industrial sector should plan the two kinds of policies as a whole… “   I have an idea what the authors are trying to say, but the grammar in this sentence is awful.  It can not go to press like this.  Could the editorial staff assist in helping the authors fix this sentence?  It is way too important to leave doubt about its meaning. 

Recommendation: Publish with minor revisions           

Author Response

Dear Reviewers,

Thank you very much for your kind letter and helpful comments, which are highly conducive to improve our paper with the number water-817698. We have thoroughly considered all comments and substantially revised our manuscript, and tried our utmost to address all concerns. As non-native speakers in English, we also invite Thomas A. Gavin, Professor Emeritus, Cornell University, for help with editing this paper. If any questions, please contact us without hesitate. Following is the revision notes for each comment.

The authors set out to examine two paths firms can use in order to reduce water pollution from their operations. The paths are a) reduce the amount of water used in the production process and b) treat the emissions flowing out of their facilities.  This is an important topic and one that is appropriate for this journal.

The authors include an introduction section where they lay out the framework and organization of their manuscript. In this section they also explain how this paper adds to the literature on the topic at hand. A key concept they lay out in this section is the concept of shadow prices, which is very important in environmental economics where amenities are not directly bought and sold on a market, and therefore lack explicit, observable prices, but has costs and benefits nonetheless and can therefore only be analyzed economically via shadow prices.

  1. In section 3 the authors go into method. I really like the model the authors employ here, which is a constrained optimization problem using a LaGrangian equation. This is highly appropriate. I do want to say however, that there is a typographical error which must be corrected. In the first order conditions, the set of equations listed as (7) the second equation in the right hand column is listed as partial F over partial W, which is a repeat of the equation in the left hand column. That equation is meant to read partial F over partial E. Please replace that W with the E. Other than that, the conditions are all correctly derived.

Responses: Many thanks for your kind reminder, and we revised the errors in equation (7).

  1. In the results section, in the text, the authors present 2015 shadow prices on water reduction per ton in various regions of China. They also use figures and a table to show how the shadow prices have changed over the period 2004-2015. They included a discussion section which focused on efficiency.

In the conclusion, the authors do a good job of summarizing the implications of their work.  However, the sentence in the final paragraph on implications that begins with 1) is not clear. It reads: “public policies that aim to save water resources closely related to emissions reduction, and then the industrial sector should plan the two kinds of policies as a whole… “I have an idea what the authors are trying to say, but the grammar in this sentence is awful.  It cannot go to press like this. Could the editorial staff assist in helping the authors fix this sentence?  It is way too important to leave doubt about its meaning.

Responses: We revised this sentence to make it clear as the following.

“1) Industrial sector can reduce wastewater emissions either by saving water use, or by treating wastewater emissions, and then policies aiming at reducing the emissions should consider the two kinds of paths as a whole.”
